# Modeling *Schistosoma japonicum* Infection under Pure Specification Bias: Impact of Environmental Drivers of Infection

**DOI:** 10.3390/ijerph16020176

**Published:** 2019-01-09

**Authors:** Andrea L. Araujo Navas, Frank Osei, Lydia R. Leonardo, Ricardo J. Soares Magalhães, Alfred Stein

**Affiliations:** 1Faculty of Geo-information Science and Earth Observation (ITC), University of Twente, PO Box 217, 7500 AE Enschede, The Netherlands; f.b.osei@utwente.nl (F.O.); a.stein@utwente.nl (A.S.); 2Institute of Biology, College of Science, University of the Philippines Diliman, UP Diliman 1101 Quezon City, Philippines; lrleonardo@up.edu.ph; 3UQ Spatial Epidemiology Laboratory, School of Veterinary Science, The University of Queensland, Gatton 4343 QLD, Australia; r.magalhaes@uq.edu.au; 4Child Health and Environment Program, Child Health Research Centre, The University of Queensland, South Brisbane 4101 QLD, Australia

**Keywords:** schistosomiasis, pure specification bias, uncertainty, Bayesian statistics, convolution model

## Abstract

Uncertainties in spatial modeling studies of schistosomiasis (SCH) are relevant for the reliable identification of at-risk populations. Ecological fallacy occurs when ecological or group-level analyses, such as spatial aggregations at a specific administrative level, are carried out for an individual-level inference. This could lead to the unreliable identification of at-risk populations, and consequently to fallacies in the drugs’ allocation strategies and their cost-effectiveness. A specific form of ecological fallacy is pure specification bias. The present research aims to quantify its effect on the parameter estimates of various environmental covariates used as drivers for SCH infection. This is done by (i) using a spatial convolution model that removes pure specification bias, (ii) estimating group and individual-level covariate regression parameters, and (iii) quantifying the difference between the parameter estimates and the predicted disease outcomes from the convolution and ecological models. We modeled the prevalence of *Schistosoma japonicum* using group-level health outcome data, and city-level environmental data as a proxy for individual-level exposure. We included environmental data such as water and vegetation indexes, distance to water bodies, day and night land surface temperature, and elevation. We estimated and compared the convolution and ecological model parameter estimates using Bayesian statistics. Covariate parameter estimates from the convolution and ecological models differed between 0.03 for the nearest distance to water bodies (NDWB), and 0.28 for the normalized difference water index (NDWI). The convolution model presented lower uncertainties in most of the parameter estimates, except for NDWB. High differences in uncertainty were found in night land surface temperature (0.23) and elevation (0.13). No significant differences were found between the predicted values and their uncertainties from both models. The proposed convolution model is able to correct for a pure specification bias by presenting less uncertain parameter estimates. It shows a good predictive performance for the mean prevalence values and for a positive number of infected people. Further research is needed to better understand the spatial extent and support of analysis to reliably explore the role of environmental variables.

## 1. Introduction

Schistosomiasis (SCH) is a water-borne infection caused by parasitic worms known as schistosomes. People get infected by skin penetration of the infective stage of the parasite. Three schistosomes species cause the infection: *Schistosoma mansoni*, *Schistosoma japonicum*, and *Schistosoma haematobium*. Among these, *S. japonicum* is the hardest one to control due to its zoonotic life cycle [1], which includes the infection of an amphibious snail from the species *Oncomelania hupensis quadrasi* as the intermediate host, and humans and other mammals as definitive hosts [2,3]. Schistosomiasis is a disease of public health significance [4,5] since it affects more than 252 million people worldwide [6]. This especially concerns communities in tropical and subtropical areas, where access to clean water and sanitation is limited. Schistosomiasis leads to malnutrition, which causes anemia and stunted growth in school-aged children [7,8]. 

Schistosomiasis risk mapping has enabled the identification of at-risk populations to target mass drug administration campaigns for disease control [9]. Mapping SCH involves the use of geographic information systems, remote sensing, and global positioning systems (GPS). These help to allocate data about infection and the physical and biological environmental variables in space. Environmental variables together with various statistical methods have been combined to model the distribution of the disease [10,11,12,13] and to quantify the role of the environmental exposure factors on SCH risk [14].

Spatial epidemiological studies are susceptible to uncertainties, which are inherent in spatial information [14,15,16]. Most of these uncertainties are caused by positional measurement errors due to GPS inaccuracies, multiple addresses, geocoding errors, misalignments between covariates and disease outcomes, and disease outcome or covariate aggregations [14]. Particularly, disease outcome and covariate aggregations at a specific administrative level could be incorrectly obtained for individual-level inference [17]. Health data are often available at a specific administrative level (i.e., group-level) while environmental data consists of a set of recorded values aggregated at monitor sites or gridded data derived from remote sensing images [18]. Spatial aggregation occurs due to the lack of geolocated information at the individual level, caused by the scarcity of sampling resources, availability of associated data or the need to protect confidentiality [14]. 

Disease outcome and covariate aggregations cause an ecological bias or ecological fallacy. This represents an important source of uncertainty [14,15,17,18,19] because any direct link between exposure and health outcomes is imperfectly measured [17]. For instance, Wakefield and Lyons [20] mention that the fundamental problem of this kind of spatial aggregations is the loss of information. Thus, the function used in the regression modeling does not represent the real relationship between the affected population and their exposure. This type of ecological fallacy is also called pure specification bias and arises due to the loss of information when a non-linear model changes its form under aggregation [20,21]. It is called ‘pure’ because it specifically addresses a model specification bias [21].

Several efforts have been made to address pure specification bias resulting in disaggregation methods. For instance, Prentice and Sheppard [22] suggest an ‘aggregated data’ method to create models based on exposure information available for a subset of individuals. Richardson et al. [23] assume parametric distributions for within-area exposures to derive accurate risk functions. Wakefield and Shaddick [18] propose a convolution model and derive an appropriate likelihood function for a scenario where health outcome data are aggregated at the district level, and exposure information is known at monitoring sites. Wang et al. [24] address pure specification bias in the least informative data scenario for aggregated disease counts with associated counts of the population at-risk, and a separate set of point level exposures from monitoring stations. They propose a conceptual probability of the incidence surface over the entire study region as a function of an exposure surface. This probability surface was then used to simulate individual disease outcomes and to obtain individual-level parameter estimates.

For a tropical disease such as SCH, the availability of individual-level infection data obtained from schools or health care centers is common [25,26,27,28,29]. Nevertheless, there are several SCH epidemiological studies [9,30,31,32,33] that limit their modeling to the outcome and covariate data aggregated at a specific administrative level (i.e., ward, municipal, province, county, district, barangays, among others) without taking pure specification bias into consideration. Moreover, there are no up-to-date studies that have quantified the effect of the covariate information on the parameter estimates when accounting for pure specification bias in SCH modeling. 

The objective of this study is to quantify the effect of pure specification bias on the parameter estimates of various environmental covariates used as drivers for SCH infection. To achieve this objective we aim to: (i) use a spatial convolution model that removes pure specification bias by using group-level health outcome data and individual-level environmental covariate data, (ii) estimate group and individual-level covariate regression parameters, and (iii) quantify the difference between the parameter estimates and predicted disease outcomes from the convolution and ecological models.

## 2. Materials and Methods 

### 2.1. Data on Human *Schistosoma japonicum* Infection, Study Area, and Sampling Design

We use *S. japonicum* infection data collected as part of the 2008 Nationwide Schistosomiasis Survey in The Philippines [34]. In this case, *S. japonicum* is endemic in 28 of its 81 provinces [35], with approximately 1.8 million estimated infected people [36]. The disease affects children, adolescents, and individuals with high-risk occupations, such as farmers and fishermen [36,37]. 

A two-stage systematic cluster sampling was used in the sampling design. Stratification was done by a prevalence level (high, medium, and low), obtained from the 1994 World Bank-assisted Philippine Health Development Program. Provinces and barangays were the primary and secondary sampling units respectively. A barangay is the smallest administrative division in the Philippines, numbering about 22–50 in a single municipality. Provinces with high and moderate prevalence rates were included, while the random selection was done among the low-prevalence provinces and non-endemic provinces. Within the selected provinces, high prevalence barangays were also selected. We decided to work in the Mindanao region due to the good spatial coverage of the sampling over the whole region (Figure 1), and the high response rate of 70 percent [9,34,38]. In total, 22 provinces were surveyed, and between 2 and 10 barangays were surveyed per province. In total, 108 out of 10,021 barangays were surveyed in Mindanao.

Data from 19,763 individuals were recorded in the survey but not georeferenced. Information regarding the corresponding barangay and province were recorded for each individual. For this reason, individual-level survey data were aggregated and geo-located to the centroids of 108 barangays in Mindanao. We used a probability of infection p in barangay k as our disease outcome variable.

Kato-Katz thick smear examination [39] was used to diagnose *S. japonicum* infection based on two-sample stool collection. Each sample was read using a microscope and the presence of *S. japonicum* eggs indicated active infection. Due to inconsistencies in the submission of the second stool sample, however, only the results of the first stool sample were available [9] from people aged two years and above. Information such as gender and age were recorded for each individual. More details about the sampling design can be found in Leonardo et al. [34,38].

### 2.2. Environmental and Geographical Data

Six environmental variables were included in the analysis: the nearest distance to water bodies (NDWB), normalized difference vegetation index (NDVI), normalized difference water index (NDWI), day (LSTD) and night (LSTN) land surface temperature, and elevation (E). The nearest distance to water bodies was calculated using the closest facility network analysis tool in ArcGIS version 10 [40]. We used rivers and lakes as water bodies. As input for the network, we used the river and road networks, and the location of cities and hamlets. We first calculated the NDWB for each city point and then interpolated the distance values for all the surveyed barangays. Interpolation was performed using Ordinary Kriging. The nearest distance to water bodies shows the accessibility of people to water bodies since they represent the main infection foci. We used NDVI obtained from the Modis MOD13Q1 product. The normalized difference vegetation index served as an indicator of vegetation presence and greenness, particularly the presence of flooded agricultural land such as paddy fields, which is an important factor for Asian schistosomiasis [41]. We included NDWI from the Google Earth Engine as an annual Landsat 7 composite for the year 2008. The normalized difference water index was used as a proxy indicator of flooding [42,43]. The day land surface temperature was also included and obtained from the Modis MOD11A2_LST product. The day land surface temperature is determinant for the survival of larval stages of snails [44,45,46] and is used as a proxy for water temperature given that the thermal condition of shallow waters usually reflects the ambient temperature of the air [9]. Elevation was also included and obtained from ASTER GDEM version 2 from United States Geological Survey (USGS) [47]. In the Philippines, the presence of snails is also driven by the local topography [48,49,50]. At lower altitudes, the risk of finding snails increases. Table 1 summarizes the information about environmental information and their sources.

### 2.3. Convolution Model (Individual-Level Model)

An individual is considered infected if at least one parasite egg is found. *S. japonicum* infection data y are available at individual-level i recorded within a barangay k. Various n environmental variables x are available for each image pixel. Because the exact response locations of the $y_{ik}$ are unknown, individual-level data are aggregated to their corresponding barangay centroid and are, thus, denoted by $y_{k}$. A naive group-level model is given by Equation (1).
(1)$$y_{k}|{\bar{x}}_{k},\gamma \sim Binomial\left(N_{k},{\hat{p}}_{k}\right)$$
where $N_{k}$ and $p_{k}$ are the number of sampled individuals and the probability of infection in barangay k, respectively, and ${\bar{x}}_{k}$ is the observed mean exposure within barangay k. The probability ${\hat{p}}_{k}$ is modelled based on Equation (2) using environmental variables as predictors, where γ are the group-level covariate coefficients.
(2)$$logit({\hat{p}}_{k})=\;\gamma_{0}+\gamma_{1}\cdot {{\bar{x}}_{1}}_{k}+\gamma_{2}\cdot {{\bar{x}}_{2}}_{k}+\ldots +\gamma_{n}\cdot {{\bar{x}}_{n}}_{k}\;$$

We suppose that, for an individual i in area k, $y_{ik}$ follows a Bernoulli distribution (Equation (3)). Then an individual level model is presented in Equation (4), where the parameters β are the individual-level coefficients. However, this model assumes that we know the individual level locations.
(3)$$y_{ik}|x_{ik},\beta \sim Bernoulli\left(p_{ik}\right)$$
(4)$$logit(p_{ik})=\;\beta_{0}+\beta_{1}\cdot {x_{1}}_{ik}+\beta_{2}\cdot {x_{2}}_{ik}+\ldots +\beta_{n}\cdot {x_{n}}_{ik}$$

Pure specification bias will result in γ
≠ β, where the relationship between aggregated disease risk and exposure on areal units differs from the relationship between the individual disease risk and the associated exposure. In a non-linear model, as in our case, this difference is produced by a loss of information due to aggregation known as pure specification bias. Pure specification bias is reduced in size since the ‘within area’ exposure is more homogenous [20]. This could be obtained by having a finer partition of space in which exposure measurements are available [18,20]. 

As we know the individual-level responses $y_{ik}$ but not their locations, we minimized the pure specification bias by extracting covariate information from cities within the barangays. We selected cities since they were the finest units we found available by taking them as a proxy for exposure locations at an individual level. Cities were extracted from the 2010 build-up data base from the National Mapping and Resource Information Authority from The Philippines [51]. Cities were not available for all the surveyed barangays. Therefore, we digitalized them using Google Earth images.

We used the aggregate data method proposed by Prentice and Sheppard, 2001 [22]. Let exposure or covariate data $x_{jk}$ be measured at locations $s_{jk}$
$j=1,\ldots ,m_{k}\leq N_{k}$, for a subset of individuals. Then, we estimated the average risk of the individuals in area k, and the individual level coefficients β. This was done by calculating the mean of the risk function (Equation (5)) instead of evaluating the risk function at the mean exposure (Equation (2)). In this way, the average ${\hat{\hat{p}}}_{k}$ of the function over the exposures corrects for the pure specification bias and differs from the function evaluated at the average exposure $\left({\hat{p}}_{k}\right)$. Thus, ${\hat{\hat{p}}}_{k}$ is the estimated average probability of infection of the individuals in area k.
(5)$${\hat{\hat{p}}}_{k}=\frac{1}{m_{k}}\cdot \overset{m_{k}}{\underset{j=1}{\sum }}\frac{1}{1+\exp(-\left(\beta_{0}+\beta_{1}\cdot {{\bar{x}}_{1}}_{jk}+\beta_{2}\cdot {{\bar{x}}_{2}}_{jk}+\ldots +\beta_{n}\cdot {{\bar{x}}_{n}}_{jk}\right))}\;$$

In our study, covariate data were obtained at a finer level of analysis than the barangay. Therefore, we assumed that averaged covariate values at city level j represent individual covariate values (i.e., $x_{jk}=\;x_{ik}$). We then know $x_{jk}$ but not the geographical linkage with individuals. One way to account for this is to allocate $N_{jk}$ individuals to measurement $x_{jk}$ by equally dividing the population. Therefore, $N_{jk}=N_{k}/m_{k}$, is a simple version of the convolution model. 

The spatial convolution model is represented in Equation (6), where the risk function also accounts for the spatial variability by including spatial ($s_{k}$) structured random effects.
(6)$${\hat{\hat{p}}}_{k}=\frac{1}{m_{k}}\cdot \overset{m_{k}}{\underset{j=1}{\sum }}\frac{1}{1+\exp(-\left(\beta_{0}+\beta_{1}\cdot {{\bar{x}}_{1}}_{jk}+\beta_{2}\cdot {{\bar{x}}_{2}}_{jk}+\ldots +\beta_{n}\cdot {{\bar{x}}_{n}}_{jk}+s_{k}\right))}\;$$

The implemented convolution model that corrects for the ecological fallacy is of the form.
(7)$$y_{k}|{\bar{x}}_{jk},\;\beta \sim Binomial\left(N_{k},{\hat{\hat{p}}}_{k}\right)$$

The model includes an intercept ($\beta_{0}$), averaged city-level environmental variables (${\bar{x}}_{jk}=NDVI,NDWI,LSTD,LSTN,E,NDWB$), and their corresponding individual-level coefficients β, and a spatial random effect ($s_{k}$) as described in Equation (6). All covariates were standardized to have mean = 0 and standard deviation = 1. Collinearity between covariates was assessed with the Pearson correlation coefficient.

Prior information for the intercept $\beta_{0}$ was given as a proper uniform distribution with wide bounds (−100, 100). The other β parameters were given non-informative normal distributions with mean = 0 and precision 1/$\sigma^{2}$, with σ uniformly distributed on a wide range (0, 100). These distributions are recommended in order to avoid overestimations on the parameters [52]. These parametrizations allow a good mixing of the sequences used for Marcov Chain Monte Carlo simulations and contribute to their faster convergence [53]. 

Spatial dependence was modelled using a spatially structured random effects distribution based upon a geo-statistical model. This model can be used as a sampling distribution for continuous spatial data [54]. The vector of random variables s associated with point locations ($x_{k},y_{k}$), k=1,…,K, was modelled with a multivariate normal distribution $s\sim MVN_{K}\left(\mu ,\;\Sigma_{ab}\right)$ with mean μ = 0 and a covariance matrix $\Sigma_{ab}$ defined by a powered exponential spatial correlation function from Equation (8).
(8)$$\Sigma_{ab}=\sigma^{2}\cdot \exp\left[-{\left(\phi \cdot d_{ab}\right)}^{\kappa }\right]$$

The covariance matrix is specified as a function of the distances $d_{ab}$ between barangay centroids a and b, the rate of decline of spatial correlation per unit of distance ϕ, the scalar parameter representing the overall variance $\sigma^{2}$, and the scalar parameter κ controlling the amount of spatial smoothing. Since it is often difficult to learn much about the κ parameter, and large values of κ could lead to smoothing, we used κ=1. The prior distribution for ϕ was set uniform with lower and upper bounds at $2\times 10^{-7}$ and $3\times 10^{-3}$, respectively. These values give a diffuse but plausible prior range of correlations between 0.1 and 0.99 at the minimum distance between points (575 m) and between 0 and 0.3 at the maximum distance between points (<552 km), which assists identification [55]. The variance parameter was given a half-normal distribution with mean = 0 and variance = 1. Half-normal was selected in order to restrict our prior to positive values and avoid problems with convergence [52,56].

The model was run using three sequences or chains with 50,000 iterations that ensured simulations representative of target distributions [53] and a good stability for convergence [53]. A burn-in of 25,000 iterations was used, discarding the first half of each sequence that is used to diminish the influence of starting values [53]. We monitored convergence visually and statistically, first by inspecting at the trace plots and then by checking the $\hat{R\;}$ statistic [57,58], which are also called a potential scale reduction factor. This assesses sequences mixing by comparing the between and within variation. $\hat{R}$ values < 1.1 indicate evidence that sequences had converged [57], while high values suggest that an increase in the number of simulations may improve our inferences [53]. Data were structured in a rectangular format, where the columns are headed by the array name. The survey data and the codes in bugs for the convolution and ecological models are provided in the supplementary files 1 and 2, respectively.

### 2.4. Ecological Model (Group-Level Model)

As a comparison, we estimated the group-level covariate regression parameters γ by using Equations (1) and (2). We used ${\hat{p}}_{k}$ as a function of the same covariate information, ${\bar{x}}_{k}=NDVI,NDWI,LSTD,LSTN,E,NDWB$, but averaged for each surveyed barangay, and added the spatial random effects term $s_{k}$ (Equation (9)). Prior distributions for γ and $s_{k}$ were the same as the ones given for the β parameters and the geo-statistical spatial term.

(9)$${\hat{p}}_{k}=\overset{K}{\underset{k=1}{\sum }}\frac{1}{1+\exp(-\left(\gamma_{0}+\gamma_{1}\cdot {{\bar{x}}_{1}}_{k}+\gamma_{2}\cdot {{\bar{x}}_{2}}_{k}+\ldots +\gamma_{n}\cdot {{\bar{x}}_{n}}_{k}+s_{k}\right))}$$

We compared estimated covariate regression coefficients and their credible intervals from both models, and also the data generated from the posterior predictive distribution to the observed data. Posterior predictive distributions were generated using simulations. Residuals were calculated by subtracting the simulated values from the observed values. We created correlation and residual plots for convolution and ecological models for three different simulations. Lastly, we compared the predicted prevalence values for both models and their corresponding credible intervals for each barangay.

### 2.5. Model Validation

In order to assess the model fit, we compared the deviance information criterion (DIC) values between simple and spatial models from the convolution and ecological models. Convolution and ecological models were validated using two methods. First, we used the posterior predictive distribution to check our model assumptions by comparing the data generated from the simulations of the predictive distribution to the observed data using a test statistic. The test statistic generates a posterior predictive *p*-value (pp*p*-value) by calculating the proportion of the predicted values that are more extreme for the test statistic than the observed value for that statistic. If the data violated our model assumptions, the observed test statistic should differ from most of the replicated test statistics from our model (i.e., pp*p*-value close to 0 or 1). If the model fits the data, a pp*p*-value around 0.5 is expected. Test statistics and pp*p*-values were created for maximum, minimum, and mean values for both, convolution, and ecological models. 

Second, we used the area under the curve (AUC) of the receiving operating characteristics (ROC) for applying a threshold of 0.5%, which is the prevalence mean in the Mindanao region. Thus, we would like to know the ability of the model to discriminate the mean prevalence level in the study area. We also investigated the ability of the models to discriminate the number of positive cases. Thus, a threshold of 1 was used, which indicates the presence of at least one positive case. An AUC value of 70% was taken to indicate acceptable predictive performance [9,59].

### 2.6. Software and Data Sources

Barangay centroids were obtained from an up-to-date barangay shape file from a DIVA geographic information system [60]. River and road networks were obtained from the Open Street Map Project in the Philippines [61]. Locations of cities and hamlets were extracted from the National Mapping and Resource Information Authority from The Philippines [62] from 2010.

The model was implemented in the software OpenBUGS 3.2.3 [63,64] (Medical Research Council, Cambridge, UK and Imperial College London, UK). The software is available for free at [65]. We used the package R2OpenBUGS [66] to call OpenBUGS from R. The spatial models were coded using functions from GeoBUGS [55], which is an add-on module to OpenBUGS that provides an interface to work with conditional autoregressive and geo-statistical models. Data pre-processing and Ordinary Kriging was performed in R [67]. 

### 2.7. Ethics Approval

The data used in this study was collected in 2005 when there was no requirement for ethical review and clearance. This study used aggregated (barangay-level) survey data, which enabled full de-identification of individuals involved in the survey.

## 3. Results

### 3.1. Convolution Model 

Posterior means and credible intervals resulting from the simple version of the convolution model (Equation (5)) are given in Table 2. The credible intervals did not include zero values, which shows that all covariates have a strong effect in the observed outcomes except NDVI. We decided to discard NDVI from the spatial convolution and ecological model (Equation (6)). Deviance information criterion values for the simple and spatial models were 419.8 and 64.13, respectively. This shows that the spatial model performs better than the simple model. This is also supported by the residual spatial variation of the survey data presented in Mindanao [9]. Table 2 shows the resulting parameter estimates and credible intervals for the spatial convolution model.

### 3.2. Ecological Model 

As in the convolution model, credible intervals resulting from the simple version of the ecological model (Equation (2)) showed that all covariates have a strong effect in the observed outcomes except NDVI. The normalized difference vegetation index was also discarded from the spatial ecological model (Equation (9)). Deviance information criterion values for the simple and spatial models were 388.1 and 126.4, respectively. This shows that the spatial model is more adequate. Table 2 shows the resulting parameter estimates and credible intervals for the spatial ecological model.

### 3.3. Convolution Versus Ecological Model

Figure 2 shows the resulting density plots from the regression parameter estimates derived from the ecological and convolution models. For the intercept parameter (Figure 2a), the convolution model estimated a lower mean value (difference = 0.11) (Table 2) with a higher uncertainty, or credible interval width, than the ecological model (Table 2). The regression parameter estimate for NDWI from the convolution model shows a higher mean value than the one from the ecological model (Table 2 and Figure 2b). The difference between these estimates is around 0.28 with the same uncertainty for both models (Table 2). In the case of LSTD (Figure 2c), the estimated mean values from the convolution model are higher but with a slightly lower uncertainty than the ecological model. Difference between estimates are around 0.16 (Table 2). For the LSTN parameter (Table 2) and Figure 2d, the convolution model estimated a lower mean value with lower uncertainty than the ecological model (Table 2). The difference between these estimates is approximately 0.2. In the case of the elevation parameter (Figure 2e), estimated mean values from the convolution model are slightly higher than estimates from the ecological model (difference = 0.08) (Table 2) and have lower uncertainty than the ecological model estimates (Table 2). The estimated parameter value for NDWB (Figure 2f) is lower in the convolution than in the ecological model (Table 2). The difference in this value is relatively small at around 0.03. Nevertheless, uncertainty is higher in the convolution than in the ecological model (Table 2).

Differences between observed and predicted prevalence values are similar for both models ($R^{2}$ = 0.9). For the convolution and ecological model, the maximum difference between the predicted and the observed prevalence values is around 1% (Figure 3a,b). Figure 3a,b show that, for fitted prevalence values higher than 2%, both models underestimate the prevalence of infection, while, for fitted prevalence values lower than 2%, both overestimation and underestimation occur. Both models show similar predicted values with a maximum difference of 0.3%. The uncertainty in the predictions from both models is the same in 88 barangays. The ecological model presents a higher uncertainty than the convolution model in 9 barangays, while the convolution model shows higher uncertainty than the ecological model in 11 barangays.

### 3.4. Model Validation 

The maximum and minimum observed prevalence values are 0.085 and 0, respectively. For the convolution model, the pp*p*-values for the maximum and minimum observed values are 0.65 and 1, respectively. This shows that it is likely to see the maximum and minimum prevalence values from the observed data in the predicted data. The highest pp*p*-value of 1 assures that 100% of our predicted data contain the minimum observed value. This could be due to an over fit to the data for small prevalence values. For the ecological model, the maximum and minimum pp*p*-values are 0.67 and 1, respectively. Like the convolution model, it is likely to see the maximum and minimum observed prevalence values in our predicted data and there might be an over fit to the data for small prevalence values.

Results from the second validation method show the high ability of both models to predict prevalence values with an AUC equal to 93% and 94% for the convolution and ecological models, respectively. The AUC values with respect to predictive ability of the number of cases are 81% and 94% for the convolution and ecological models, respectively. Both models can discriminate the number of positive cases of schistosomiasis and mean prevalence values. Nevertheless, as we can see, the ecological model might over fit the data as compared to the convolution model with respect to the number of positive cases. 

## 4. Discussion

Several studies have modelled SCH disease risk using surveyed and environmental aggregated information at an administrative-level [9,30,31,32,33]. These studies so far ignored the pure specification bias caused by the use of ecological or group-level estimates as individual-level estimates. Only a few studies [18,24] have quantified the influence of pure specification bias on the regression parameter estimates and all studies on SCH ignored the influence of pure specification bias on disease predictions. In our paper, we quantified the effect of pure specification bias on assessing the parameters for environmental covariates that are used for the mapping of *S. japonicum* infection risk. Our contribution is both methodological and practical.

Our starting point was that NDVI, NDWI, LSTD, LSTN, elevation, and NDWB are relevant for SCH transmission [68,69,70]. For instance, NDVI is an indicator of flooded vegetation [9], specifically rice paddy fields, and environmental moisture [42,71]. In both models, all variables have a strong effect in the observed range of outcomes, except NDVI. An explanation could be the effect of spatial support of this variable at 250 m. The International Rice Research Institute (IRRI) estimated an area substantially smaller than 25 ha for rice paddy fields [72]. The area covered by an NDVI pixel equals 6.25 ha. This shows that a spatial support of 250 m is still too coarse to reliably represent paddy fields. It could be relevant to use a higher spatial support to reliably assess the role of NDVI. For instance, Walz et al. [42] have successfully delineated paddy fields by using NDVI from RapidEye imagery at a higher spatial support of 5 m.

In the case of NDWI, the convolution model estimates a higher NDWI parameter value than the ecological model, but closer to zero (Figure 2b). Hence, it may not have a strong effect on the observed range of outcomes. The difference between convolution and ecological models when estimating the NDWI parameter is high (0.28) as compared to the differences for other variables. This could be due to (i) the decrease of the spatial extent of analysis from barangay to a city-level for covariate extraction, and (ii) the coarse spatial support of the variable at 500 m. A correspondence between a decrease in the extent of analysis and the within cities variability of NDWI values would change the average value used for parameter estimation. This could yield the high differences in the estimates and, together with the coarse support of the variable, could lead to a weaker effect of NDWI in SCH prevalence. Hence, NDWI pixels of 0.25 km^2^ are too coarse to reliably define flooded zones in city areas that range from 0.02 to 3 km^2^. The uncertainty in NDWI parameter estimate is similar for both models, as shown by the credible interval with of 0.44 (Figure 2b and Table 2). A possible explanation is that NDWI values are similar between and within cities, and between and within barangays. For instance, NDWI values between cities and between barangays range from 0.095 to 0.51 and from 0.099–0.51, respectively. The average NDWI value within a specific barangay is 0.3, while NDVI values range from 0.29–0.31 for the cities it contains.

For LSTD, a higher estimated value in the convolution model was observed than in the ecological model (Figure 2c). As for NDWI, this could be (i) due to the decrease in the within cities variability corresponding to a decrease in spatial extent for covariate values extraction, and (ii) the coarse LSTD spatial support of 1 km. The day land surface temperature, area pixels of 1 km^2^ are too coarse to reliably define low and high-temperature zones in city areas ranging from 0.02 to 3 km^2^. The uncertainty is slightly lower in the convolution model than in the ecological model possibly due to the similarity of the LSTD variability between and within cities and between and within barangays. For instance, similar LSTD values ranging from 25 to 35.7 °C and from 21.7 to 36 °C were found for the convolution and ecological models, respectively.

A different pattern is observed for the estimate of the LSTN parameter. In contrast to LSTD, the estimate and uncertainty from the convolution model are lower than for the ecological one (Figure 2d). This could be explained by the difference between the LSTN variability within cities and within barangays. For instance, we selected a barangay bigger in size than the cities it contains. We compared LSTN values from the cities and from the barangay and found differences from 4 to 7 °C, although the spatial support of LSTN is coarse at 1 km. These differences are small, but could be determinant for the parasite presence, as the distribution of SCH is driven by water temperatures from 15 to 20 °C [73]. This means that we could find the parasite in the barangay with an average LSTN value of 20 °C, but we could not find it in the cities inside the barangay, with LSTN values ranging from 22–24 °C. Thus, there is a clear loss of information produced by a pure specification bias.

Elevation presents an estimate closer to zero in the convolution model as compared to the ecological model (Figure 2e). This is possibly related to the decrease in the within cities elevation variability as a result of the decrease in the spatial extent of covariate extraction, from barangays to cities. Although its spatial support is relatively high, i.e., 30 m, within cities variability decreases because the Mindanao region does not contain steep slopes or sharp changes in elevation. Changes in elevation are gradual, which means that a city can share a single elevation value. The uncertainty in the convolution model is lower than in the ecological model. The reason could be the large difference in the between and within cities and barangays variability. For instance, elevation values between the cities ranged from 0 to 0.9 km, while elevation values between barangays ranged from 0 to 1.3 km. We also compared the elevation values averaged in a specific barangay and the range of values from the cities within the barangay. The averaged barangay value was around 1.3 km, while the city values within the barangay ranged from 0.87 to 0.89 km. These differences are high and give an idea of the large amount of information that could be lost when estimating individual parameters at ecological levels of analysis.

Lastly, for NDWB, the convolution model estimates a lower parameter value but with a higher uncertainty (Figure 2f). This could be explained by the discrepancy between the within barangay and city variability. For instance, for a specific barangay, the averaged NDWB value was approximately 3.75 km, while NDWB values for the cities within this barangay ranged from 0.4 to 6.4 km. In addition, the NDWB values between the cities ranged from 0.17 to 26.2 km, while the NDWB values between the barangays ranged from 0.26 to 15.5 km. The higher uncertainty in the convolution model could be explained by the use of ordinary Kriging in the NDWB calculation. The use of interpolation increases the variance in the estimates in a somewhat unrealistic way since it uses a constant mean [54] while, in reality, the different cities and barangays have different means.

The present research shows that the ecological and convolution models present similar prediction results. However, the proposed convolution model is preferred based on the lower uncertainties found in most of its parameter estimates, which shows that it corrects for pure specification bias. Moreover, according to our validation results, the convolution model has a high predictive ability to detect a positive number of cases (81%) and mean prevalence values (93%). From a public health perspective, the provision of regression coefficients that are less uncertain and better approach the individual-level estimates is a step forward to the desired uncertainty analysis in a schistosomiasis-modelling framework. This could be used as a decision support tool for helminth control programs. Moreover, less uncertain models and maps would avoid erroneous conclusions and decisions about the spatial distribution of schistosomiasis. Lastly, information on uncertainty regarding pure specification bias could guide mass drug administration campaigns by enhancing the assessment of the infection risk and understand potential impacts on human health [15,74].

### Suggestions

Spatial extent and support of analysis are relevant drivers for the model parameter estimates and their associated uncertainties. The choice of support may affect the pattern identified from the data and the relationship between environmental variables and SCH prevalence [75,76]. We recommend bringing all covariates to a common spatial support prior to analysis. A suggestion would be to start at e.g., 30 m and examine larger supports to more precisely quantify the role of an environmental variable in the disease modelling process [13,77].

The trend in the residual plots from both models (Figure 3) points to a dependence between the residuals and the fitted prevalence values. This dependence could be due to heteroscedasticity [78], which means that similar interactions between the variables could lead to different prevalence values. This does not represent a problem in the model parameter estimates but is an indicator that the model can be improved [79,80]. Although our aim is not about model fit, we could possibly improve our predictions by exploring other disease driven factors and include them in the model or explore the model specification. Perhaps fitting a zero-inflated binomial (ZIB) model could help improve the predictions given that our data show a high number of barangays with zero prevalence (~77%).

## 5. Conclusions

The present study proposes a convolution model that removes pure specification bias by using ecological, group-level, health outcome data and city-level, individual-level, environmental data. For most covariates, the uncertainties in the convolution model are lower than those in the ecological model.

The spatial extent of the covariates values and the spatial support or resolution of these covariates are relevant for the parameter estimates and their uncertainties. The spatial extent and support also influence the role of the covariates in SCH modelling. Why this happens should be further explored. Additionally, between-covariate and within-covariate variability resulted in similar uncertainties in both models. Conversely, differences in between and within covariate variability explain the loss of information produced by a pure specification bias, which leads to lower uncertainties in the convolution model.

Lastly, this study shows no significant differences in the predicted values from both models. Predicted values from the convolution model are as uncertain as the predicted values from the ecological model in the majority of surveyed barangays (81.5%). The convolution model, however, shows a good predictive performance for the mean prevalence values and a positive number of infected people. 

## Figures and Tables

**Figure 1 ijerph-16-00176-f001:**
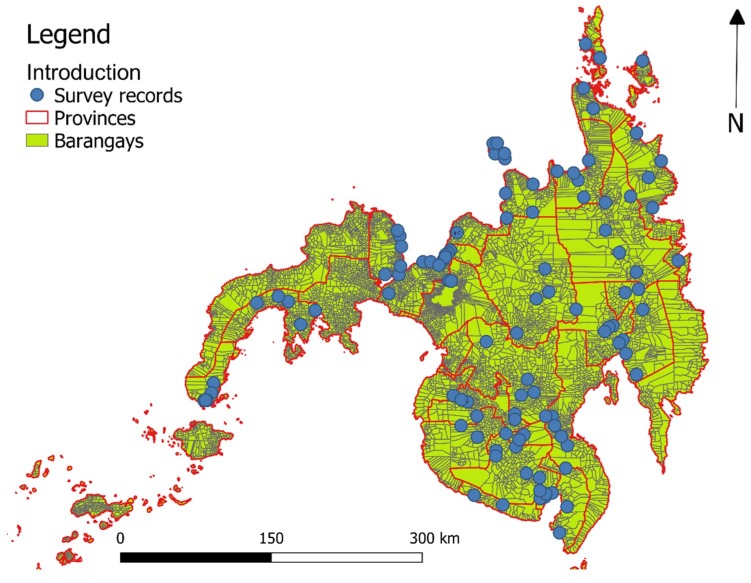
Study area.

**Figure 2 ijerph-16-00176-f002:**
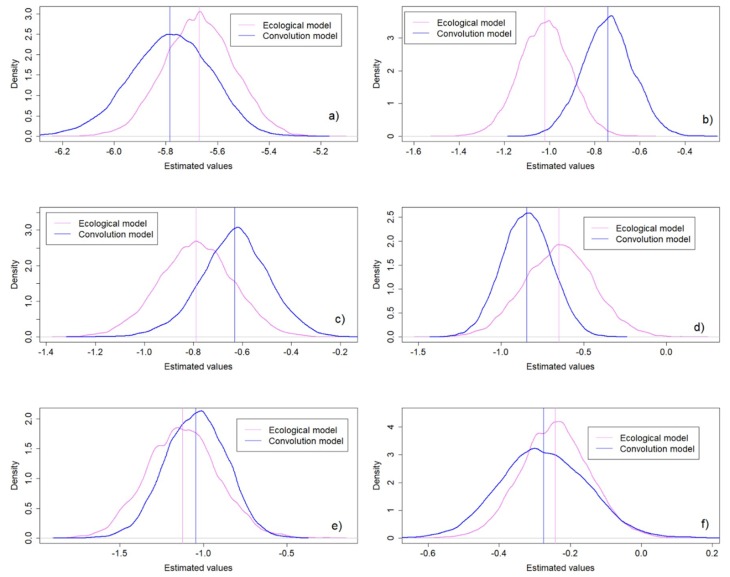
Covariate regression coefficients density plots for the convolution and ecological models. (**a**) Intercept, (**b**) Normalized Difference Water Index, (**c**) Day Land Surface Temperature, (**d**) Night Land Surface Temperature, (**e**) Elevation, and (**f**) Nearest Distance to Water Bodies.

**Figure 3 ijerph-16-00176-f003:**
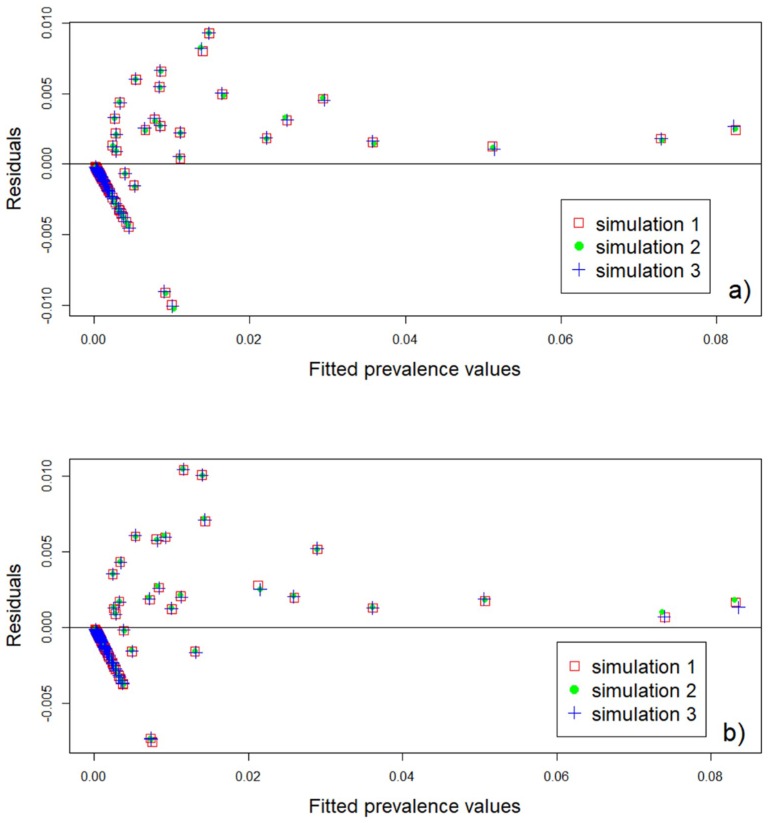
Residual plots for the (**a**) convolution and (**b**) ecological models.

**Table 1 ijerph-16-00176-t001:** Environmental variables description.

Environmental Variable	Spatial Resolution	Temporal Resolution	Data Type	Original Coordinate System	Data Source
Elevation	30 m	NA	Raster	EPSG:4326	Aster GDEM V2 from USGS
NDVI	250 m	2008	Raster	EPSG:4326	MOD13Q1
NDWI	500 m	2008	Raster	EPSG:32651	Landsat 7, one-year composite
LST	1 km	2008	Raster	EPSG:4326	MOD11A2
NDWB	250 m	2010	Raster	EPSG:32651	Derived from closest facility network using roads, urban areas, river network, and water bodies

NDVI: Normalized difference vegetation index; NDWI: normalized difference water index; LST: day land surface temperature; NDWB: nearest distance to water bodies; USGS: United States Geological Survey.

**Table 2 ijerph-16-00176-t002:** Estimated regression coefficients (mean and 95% credible intervals).

Estimated Parameters	Posterior Mean (95% Crl)		Standard Deviation		Credible Intervals Width (Uncertainty)	
	Convolution Model	Ecological Model	Convolution Model	Ecological Model	Convolution Model	Ecological Model
Intercept	−5.79 (−6.11,−5.5)	−5.67 (−5.93,−5.41)	0.16	0.13	0.62	0.53
NDWI	−0.74 (−0.96,−0.55)	−1.02 (−1.24,−0.80)	0.11	0.11	0.44	0.44
LSTD	−0.63 (−0.92,−0.38)	−0.79 (−1.08,−0.49)	0.14	0.15	0.56	0.59
LSTN	−0.84 (−1.13,−0.55)	−0.65 (−1.05,−0.24)	0.15	0.21	0.59	0.82
Elevation	−1.05 (−1.4,−0.71)	−1.13 (−1.53,−0.70)	0.18	0.22	0.71	0.84
NDWB	−0.28 (−0.51,−0.05)	−0.24 (−0.43,−0.05)	0.13	0.09	0.48	0.38
ϕ	4 × 10^−5^ (−0.004,0.004)	2 × 10^−5^ (−0.0004,0.0004)	2.00 × 10^−4^	1.00 × 10^−5^	6.80 × 10^−5^	3.50 × 10^−5^
Variance of spatial random effect	2.58 (1.7,3.6)	2.6 (1.8,3.61)	0.48	0.47	1.9	1.82

Crl: Credible interval.

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
