# Peer review of "Modeling Schistosoma japonicum Infection under Pure Specification Bias: Impact of Environmental Drivers of Infection"

_ijerph, 2019, doi:10.3390/ijerph16020176_

Round 1

Reviewer 1 Report

This manuscript tried to quantify the effect of pure specification bias on the parameter estimates of various environmental covariates used as drivers for SCH infection by using a spatial convolution model that removes pure specification bias. Their results showed that the proposed convolution model was able to correct for pure specification bias by presenting less uncertain parameter estimates, and showed a good predictive performance for the mean prevalence values and for a positive number of infected people. The article seems appropriate for publication with some changes:

Why are some environmental variables related to Schistosoma japonicum infection (e.g., rainfall) not included in these model? Please explain.

In “the nearest distance to water bodies (NDWB)”, what are water bodies? River, pool or lake? Please describe in detail.

Author Response

Point 1. Why are some environmental variables related to Schistosoma japonicum infection (e.g., rainfall) not included in these model? Please explain.

Reply 1. Thank you for your question. Rainfall was not included in the analysis because:

1)     The main source of information (Worldclim data warehouse) has monthly average data for precipitation for the years 1960-1990. We considered this as a period of time that may not represent the current time scale of analysis, which is the year 2008 when the survey was performed.

2)     Rainfall is also known as a determinant for the existence of water bodies and the survival of the larval stages of the snail [1-3]. In our study we included variables such as the normalized difference water index (NDWI) and land surface temperature (LST), which have the same application as rainfall as they are indicators of the presence of water bodies and water temperature, which is also a determinant factor in the larval stages on the parasite.

3)     Other sources of information like Meteosat 7 include rainfall information only at 8 km spatial resolution. We considered this is a very coarse resolution to extract rainfall information at city level.

Point 2. In “the nearest distance to water bodies (NDWB)”, what are water bodies? River, pool or lake Please describe in detail.

Reply 2. Thank you for your comment. The water bodies are the rivers and lakes. This is clarified in lines 153 and 154.

REFERENCES

1.           Prah, S.; James, C. The influence of physical factors on the survival and infectivity of miracidia of Schistosoma mansoni and S. haematobium I. Effect of temperature and ultra-violet light. J. Helminthol. 1977, 51 (1), 73-85,

2.           Woolhouse, M.; Chandiwana, S. Population dynamics model for Bulinus globosus, intermediate host for Schistosoma haematobium, in river habitats. Acta Trop. 1990, 47 (3), 151-160,

3.           Pietrock, M.; Marcogliese, D. J. Free-living endohelminth stages: at the mercy of environmental conditions. Trends Parasitol. 2003, 19 (7), 293-299,

Reviewer 2 Report

The manuscript by Dr. Araujo Navas and colleagues deals with an assessment of the role played by the spatial specification of environmental drivers in the modeling of Schistosoma japonicum infection probability in the Mindanao region of the Philippines. Specifically, the authors compare the outcomes of a spatial convolution model including individual-level covariates with an aggregated model based on group-level regression parameters. They found relatively minor differences in terms of the modeled output, namely schistosomiasis prevalence (delta-prev < 1 % in all cases), despite differences in the regression weights estimated in the two cases, with overall smaller uncertainties and better validation performances being associated with the spatial convolution model. The topic of the paper is interesting, in particular because the choice of resolution for a spatially explicit (epidemiological) model is typically done a priori based on matters related to data availability and numerical feasibility, and only seldom subject to any forms of ex-post evaluation. The manuscript is also quite well written and organized. All that being said, I have some comments that the authors may want to address while revising their work.

- From a work aimed at assessing possible causes of spatial fallacy, I would have expected some effort being devoted to contrasting the focal source of uncertainty, namely specification bias, against others, such as those coming from the original data. For instance, (replicated?) Kato-Katz was used as the only diagnostic tool for schistosomiasis infection, but that is far from ideal in a (relatively) low-prevalence setting like the one studied here (<<20% population infected) accroding to the WHO. How would the uncertainty associated with e.g. sub-optimal testing compare with that introduced by alternative spatial specifications of the environmental covariates?

- I am quite unsure about the framing of the comparison in terms of `convolution' vs. `ecological' models. Specifically, I do not get why the aggregated model should be termed as `ecological'. An ecological model is one that deals with ecological dynamics (including disease transmission, in this case), not one that makes specific assumptions concerning the spatial scale of analysis. While specification bias has been previously described (also) in the context of ecological models, the topic seems to be pretty general. That is why I believe that a more careful wording may help increase the visibility of the study.

Minor points

- what is the difference between equations 1 and 3?

- I cannot fully understand equation 6: why does p-hat depend on k, if you still have a summation over all k's in the rhs of the formula?

- p.8, l.336: `equations' should read `equation'

- Table 2: please double-check the credible intervals for phi

- p.11, l.390: `show to' should be `show to be' 

- is it mandatory to have temperatures expressed in K? Maybe a different scale (e.g. °C) might be easier to read for many readers

Author Response

Point 1. From a work aimed at assessing possible causes of spatial fallacy, I would have expected some effort being devoted to contrasting the focal source of uncertainty, namely specification bias, against others, such as those coming from the original data. For instance, (replicated?) Kato-Katz was used as the only diagnostic tool for schistosomiasis infection, but that is far from ideal in a (relatively) low-prevalence setting like the one studied here (<<20% population infected) according to the WHO. How would the uncertainty associated with e.g. sub-optimal testing compare with that introduced by alternative spatial specifications of the environmental covariates?

Reply 1. Thank you for your valuable comment. This study is mainly focused on one type of ecological fallacy namely: pure specification bias. This is related to the use of aggregated covariate or survey data for individual level inferences. The relationship with this uncertainty source with other sources of uncertainty, such as the diagnosis technique, is outside the scope of this research. We highlight the importance of analysing and proposing various ways to deal with uncertainties in the original data, specially the geolocation of the surveys. This is essential to accurately model the distribution of the infection.

Point 2. I am quite unsure about the framing of the comparison in terms of `convolution' vs. `ecological' models. Specifically, I do not get why the aggregated model should be termed as `ecological'. An ecological model is one that deals with ecological dynamics (including disease transmission, in this case), not one that makes specific assumptions concerning the spatial scale of analysis. While specification bias has been previously described (also) in the context of ecological models, the topic seems to be pretty general. That is why I believe that a more careful wording may help increase the visibility of the study.

Reply 2. Thank you for your valuable comment. In spatial epidemiology, ecological studies are characterized by being based on grouped or aggregated data at an specific geographical area of analysis [1-2]. Aggregated data could be covariate information only available at monitor sites, or covariate and health data available at an specific administrative level (e.g. barangay, province, municipality, etc). These aggregations may occur due to the lack of geo-located information at the individual level, caused by the scarcity of sampling resources, availability of associated data or the need to protect confidentiality [3].

The term “ecological bias” is used to describe the loss of information in the estimated associations at the individual-level produced by the use of aggregated or ecological-level information [1, 4]. This means that any direct link between health outcome data and exposure is imperfectly measured when aggregated data are used for individual-level inference [1].  

We use the term ecological model because it uses outcome and covariate information aggregated at an administrative level (i.e. barangay) and this term is appropriate according to the explanation previously stated.

Minor points

Point 3. what is the difference between equations 1 and 3?

 Reply 3. There was a mistake in equation 3. This is already corrected.

Point 4. I cannot fully understand equation 6: why does p-hat depend on k, if you still have a summation over all k's in the rhs of the formula?

Reply 4. k is an index which represents the number of barangays. The summation is given over all the cities m within an specific barangay k.

Point 5. p.8, l.336: `equations' should read `equation'

Reply 5. Corrected

Point 6.Table 2: please double-check the credible intervals for phi.

Reply 6.This is correct

Point 7. p.11, l.390: `show to' should be `show to be' 

Reply 7. Corrected

Point 8. is it mandatory to have temperatures expressed in K? Maybe a different scale (e.g. °C) might be easier to read for many readers

Reply 8. Temperatures have been changed from degrees Kelvin to degrees Celsius.

REFERENCES

1.           Wakefield, J.; Lyons, H., Spatial Aggregation and the Ecological Fallacy. In Handbook of Modern Statisticcal Methods, Hall/CRC, C., Ed. CRC Press: Boca Raton, United States, 2010; pp 541-558, ISBN 9781420072877.

2.           Wakefield, J.; Shaddick, G. Health-exposure modeling and the ecological fallacy. Biostatistics 2006, 7 (3), 438-455, doi:10.1093/biostatistics/kxj017.

3.           Zhang, Z. J.; Manjourides, J.; Cohen, T.; Hu, Y.; Jiang, Q. W. Spatial measurement errors in the field of spatial epidemiology. Int J Health Geogr 2016, 15 (21), 12, doi: 10.1186/s12942-016-0049-5.

4.           Richardson, S.; Monfort, C., Ecological correlation studies. In Spatial Epidemiology: methods and applications, Elliot, P. W., J. C.; Best, N. G.; Briggs, D. J., Ed. Oxford University Press: Oxford, United Kingdom, 2000; pp 205-220, ISBN 0192629417.

Round 2

Reviewer 2 Report

I have now read again the ms by Dr. Araujo Navas and colleagues. Unfortunately, I am left unconvinced by the answers provided by the authors, especially concerning the major points raised during the first round of review. Detailed comments follows, using the same numbering as the authors did.

1) I believe that an `out of scope' answer does not cut it, here. The authors are basically proposing that one approach (the convolution model) is better than the other (the `ecological model'). While calibration results seem indeed to suggest so (although DIC may tend to select over-fitted model, at times), validation results are much less convincing. Therefore, understanding whether sources of uncertainty other than specification bias are at play seems to be crucial, in this specific application. If the authors are not willing to investigate this issue further, at the very minimum they should motivate their choice in the ms more convincingly.

2) I see that the wording `ecological model' has been previously used with the same meaning referred to by the authors. However, I still find the choice quite diminishing (for actual ecological model[er]s) and, even more annoyingly, a direct consequence of a syllogistic fallacy (some/all ecological models are group-based; model x is group-based; model x is ecological -- a textbook example of non distributio medii). If the authors are not willing to rethink their choice of words, they should at least be adamantine in motivating what they mean by `ecological model'. Otherwise, readers with an ecological background might feel lost -- if not outright offended.

4) My bad, I meant equation 9. How does p-hat depend on k, there? I think that the main source of confusion is the use of the symbol k as dummy variable for the (outer) sum.